# Secondary Brain Injury After Parenchymal Cerebral Hemorrhage in Humans: The Role of NOX2-Mediated Oxidative Stress and Endothelin-1

**DOI:** 10.3390/ijms252313180

**Published:** 2024-12-07

**Authors:** Manuela De Michele, Paolo Amisano, Oscar G. Schiavo, Vittoria Cammisotto, Roberto Carnevale, Maurizio Forte, Vittorio Picchio, Antonio Ciacciarelli, Irene Berto, Ugo Angeloni, Silvia Pugliese, Danilo Toni, Svetlana Lorenzano

**Affiliations:** 1Stroke Unit, Umberto I Hospital, Emergency Department, Sapienza University, 00185 Rome, Italy; oscargschiavo@gmail.com (O.G.S.); antoniociacciarelli@gmail.com (A.C.); ireneberto25@gmail.com (I.B.); 2Department of Human Neurosciences, Sapienza University, 00185 Rome, Italy; paolo.amisano@uniroma1.it (P.A.); danilo.toni@uniroma1.it (D.T.); svetlana.lorenzano@uniroma1.it (S.L.); 3Department of Clinical, Internal, Anesthesiologic and Cardiovascular Sciences, Sapienza University, 00185 Rome, Italy; vittoria.cammisotto@uniroma1.it; 4Department of Medical and Surgical Sciences and Biotechnologies, Sapienza University, 04100 Latina, Italy; roberto.carnevale@uniroma1.it; 5IRCCS Neuromed, 86077 Pozzilli, Italy; maur.forte@gmail.com (M.F.); vittorio.picchio@uniroma1.it (V.P.); 6Neuroradiology Unit, Umberto I Hospital, Emergency Department, Sapienza University, 00185 Rome, Italy; ugoangeloni@yahoo.it (U.A.); s.pugliese@policlinicoumberto1.it (S.P.)

**Keywords:** intracerebral hemorrhage, oxidative stress, endothelin-1

## Abstract

Perihematomal hypoperfusion may lead to ischemic damage during intraparenchymal cerebral hemorrhage (ICH), resulting in worse prognosis. We aimed to (1) investigate the relationship between serum biomarkers related to oxidative stress and vasoactive substances and the occurrence of hypoperfusion and ischemic perihematomal lesions in ICH and (2) evaluate their correlation with the volumetric evolution of the hematoma and perihematomal edema. We enrolled 28 patients affected by ICH. Blood samples were collected at three different time points from symptom onset: T0, T1, and T2 (admission, 12–24 h, and 48–72 h, respectively), to measure endothelin-1 (ET-1), nitrites/nitrates (NO), soluble nicotinamide adenine dinucleotide 2 (NOX2)-derived peptide (sNOX2-dp), and asymmetric dimethylarginine (ADMA). Patients underwent brain MRI with perfusion study at T1 and MRI without perfusion at T2. 12 patients had ischemic perihematomal lesions at T1. A higher sNOX2-dp concentration at T0 was observed in patients with ischemic perihematomal lesions compared to those without (*p* = 0.051) and with a more severe perihematomal edema at T2 (*p* = 0.011). The ischemic perihematomal lesions development was also associated with an increased hematoma volume (*p* < 0.005), perilesional edema (*p* = 0.046), and greater midline shift (*p* = 0.036). ET-1 values at T1 were inversely correlated with hemorrhage volume at T2 (ρ = −0.717, *p* = 0.030). NOX2 activation may have a role in the development of ischemic perihematomal lesions. The association between higher ET-1 values and a lower hemorrhage volume could be related to the ET-1 vasoconstriction action on the ruptured vessel wall.

## 1. Introduction

Spontaneous intraparenchymal cerebral hemorrhage (ICH) accounts for about 20% of all strokes and is associated with high rates of mortality and long-term disability [1,2,3,4]. In recent years, besides the role of primary damage due to a direct mechanical compression by the hematoma on the surrounding tissue, many studies have highlighted the importance of the secondary damage, mostly due to the perihematomal edema and hypoperfusion, and related to oxidative stress, neuroinflammation, alteration of the blood–brain barrier (BBB), and vasoactive metabolites [5,6,7]. Hypoperfusion, together with additional factors, i.e., cerebral microangiopathy and atherosclerosis, hyperglycemia, neuroinflammation, and oxidative stress, may also be the trigger for the development of silent ischemic lesions, localized both near and in remote regions from the hematoma [8,9]. Although the pathophysiology of perihematomal ischemic lesion development is not completely ascertained, the appearance of these lesions is associated with a worse prognosis and with the recurrence of cerebrovascular events (both ischemic and hemorrhagic), cognitive decline, and death [9,10,11].

Nicotinamide adenine dinucleotide phosphate (NADPH) oxidase (NOX) is a complex multimeric enzyme that participates in the generation of superoxide or hydrogen peroxide [12,13]. NOX2 is the most represented isoform in cerebral arteries and is involved in oxidative stress-related damage [14]. Moreover, oxidative stress is a prominent cause of brain injury in ICH [15]; therefore, NOX2 could contribute to ICH. This was supported by experimental models of cerebral ischemia, which showed that NOX2 knockout mice had smaller infarcts than wild-type mice [16]. In addition, the NOX2 inhibitor apocynin improved brain edema caused by thrombolytic agents [17].

Endothelin-1 (ET-1) and nitric oxide (NO) are two vasoactive substances with opposite actions involved in the autoregulation of cerebral blood flow and neurovascular coupling [18].

ET-1 is a strong vasoconstrictor that exerts its action mainly through the ET_A_ receptors widely represented on the smooth muscle cells of brain arterioles [19]. Conversely, NO, produced by the endothelial NO synthase (NOS) isoform, acts as a powerful vasodilator [20]. There are two other NOS isoforms, one produced by neurons (nNOS) and another one produced by macrophages as a response to inflammation (inducible, iNOS). nNOS-derived NO plays a critical neuroprotective role in mediating synaptic plasticity and neuronal signaling, but it becomes a neurotoxic factor when excessive amounts of NO are produced. Therefore, in pathological conditions, this molecule acts as a double-edged sword [21].

Asymmetric dimethyl-arginine (ADMA) is an endogenous inhibitor of NOS and is known to be a marker of endothelial dysfunction and atherosclerosis. It seems to be involved in cerebral blood flow reduction and in promoting oxidative stress and inflammatory response to ischemia [22].

The main aim of this study was to assess the role of vasoactive substances (ET-1, NO, and ADMA) and oxidative stress markers (soluble NOX2-derived peptide, sNOX2-dp) in the development of perihematomal hypoperfusion and ischemic lesions. Secondly, we aimed to study the correlation between these molecular biomarkers and the evolution of hematoma and perilesional edema volume. Finally, we evaluated the impact of these variables on clinical outcome.

## 2. Results

We enrolled 28 patients with ICH, 15 males (55.2%) and 13 females (44.8%); mean (±SD) age 70.2 years (±13.8). Pre-stroke mRS was between 0 and 1 in 24 (85.2%) patients. Overall, 12 patients (42.8%) had one or more ischemic lesions in DW-MRI at T1, while there were 16 patients (57.1%) without ischemic lesions.

### 2.1. Patients’ Characteristics

We did not find any statistically significant differences between the two groups as regards demographic characteristics and NIHSS score, blood pressure, heart rate, and serum glucose values at the three assessment time points (Table 1 and Table 2). However, we found a higher prevalence of past medical history of hypertension (12 [100%] patients vs. 9 [56.3%] patients; *p* = 0.010) and of dyslipidemia (9 [75%] patients vs. 1 [6.3%] patient, *p* < 0.001) and a higher prevalence of neutrophil leukocyte counts at T0 (7.44 × 10^3^/µL vs. 4.99 × 10^3^/µL; *p* < 0.037) in the group of patients with ischemic lesions.

### 2.2. Radiological Features

Patients with ischemic lesions (Figure 1) had a median hematoma volume at T1 significantly larger than that of patients without ischemic lesions (23.05 cm^3^ vs. 6.65 cm^3^, *p* < 0.005) and a significantly higher volume of perilesional edema (15.25 vs. 8.25 cm^3^, *p* 0.046) with a more severe midline shift [4.45 mm (±4.74) vs. 1.31 mm (±2.62) (*p* = 0.036)] (Supplemental Appendix A). These data were also confirmed at T2 (Supplemental Appendix A). The “island sign” was more frequently observed in group 1 (10 [83.3%] vs. 4/15 patients [26.7%], *p* 0.003). When associations between prespecified perfusion parameters and the development of ischemic lesions at T1 or T2 were investigated, we did not find statistically significant differences.

### 2.3. Clinical Outcomes

No statistically significant between-group differences were found in terms of functional outcome at 3 months, intra-hospital death, and mortality at 3 months (Appendix A).

### 2.4. Serum Biomarkers

Patients with ischemic lesions at T1 showed a higher median serum concentration of sNOX2-dp at T0 compared to those without ischemic lesions, although the difference had a borderline statistical significance (34.9 pg/mL vs. 22.4 pg/mL, *p* = 0.051) (Figure 2 and Appendix A).

For the remaining serum biomarkers, no statistically significant differences were found between the two groups at the three time points (Appendix A).

In the exploratory multivariate analysis, after adjustment for age, sex, neutrophil absolute number at T0, and hematoma volume at T1, high sNOX2-dp values at T0 were associated with the development of perihematomal ischemic lesions at T1, although with a trend toward statistical significance (OR 1.122, 95% CI 0.991–1.269, *p*: 0.068). Changes in biomarker levels from one time point to another in the two groups are summarized in Appendix A.

### 2.5. Association Between Serum Biomarkers and Radiological Characteristics of Hematoma and Clinical Outcome

Higher levels of ET-1 at T1 significantly correlated with smaller hematoma volumes at T2 (Spearman’s rho = −0.717, *p* = 0.030) with only a trend toward the statistical significance for ET-1 levels at T0 (Spearman’s rho = −0.617, *p* = 0.077) (Figure 3).

Consistently, patients without any hematoma growth from T1 to T2 had higher levels of ET-1 at T2 compared with those with any hematoma growth (16.70 pg/mL vs. 12.83 pg/mL, *p* = 0.020) (Figure 4).

Higher levels of ET-1 at T1 were also associated with a borderline statistically significant absence of ipsilateral hemispheric hypoperfusion at T1 (17.76 pg/mL vs. 11.54 pg/mL, *p* = 0.059 (Appendix A).

Patients with a more severe perihematomal edema at T2 had higher levels of sNOX2-dp at T0 (37.99 pg/mL vs. 19.17 pg/mL, *p* = 0.011) and of NO at T2 (32.64 μM vs. 14.03 μM, *p* = 0.039) (Appendix A).

A significant linear positive correlation was found between high levels of some biomarkers and neurological severity measured by NIHSS score at different time points, such as NO at T1 (Spearman’s rho = 0.783, *p* = 0.007) with NIHSS at discharge, while there was an inverse correlation between ADMA levels at T2 and NIHSS at T2 (Spearman’s rho = −0.743, *p* = 0.009) (Appendix A).

We did not find any statistically significant association between levels of biomarkers at any time point and intra-hospital mortality or clinical outcome at 3 months measured by mRS score (Appendix A). Patients with high levels of NO at T0 (28.86 vs. 8.69 in the survivors, *p* = 0.035) were more likely to die at 3 months (Appendix A).

## 3. Materials and Methods

### 3.1. Patients

In this study, we included patients with spontaneous ICH, admitted to the emergency department of our hospital, within 6 h of symptom onset, between January 2019 and July 2021. For each patient we collected demographic data, pre-stroke modified Rankin Scale (mRS), vascular risk factors, previous medical therapy, presence of infections in the previous two weeks, and stressful factors at the time of the event. We defined three time points from symptom onset for the collection of clinical and diagnostic data: time 0, at admission (T0); time 1 (T1) at 12–24 h; and time 2 (T2) at 48–72 h. At T0, we performed a general and neurological examination, measurement of vital parameters (blood pressure, heart rate, and body temperature), and serum glucose assessment, and we addressed stroke severity by using the National Institutes of Health Stroke Scale (NIHSS) score and a plain brain computed tomography (CT) scan. We took blood samples at the three prespecified time points for the routine blood tests and for dosing the following molecular biomarkers: ET-1, NO, sNOX2-dp, and ADMA as described below. The T0 sampling was collected prior to the administration of any drug. At T1, we performed a brain magnetic resonance imaging (MRI) scan, including diffusion (DW) and perfusion (PW) weighted images, clinical evaluation with NIHSS, and blood pressure measurement. At T2, we undertook a follow-up MRI (DWI), an apparent diffusion coefficient (ADC) map, a fluid-attenuated inversion recovery (FLAIR), and a gradient echo (GE).

Finally, we evaluated the level of disability at 3 months after the acute event using the modified Rankin scale (mRS) by telephone interview (T3). A schematic representation of the study’s experimental design is depicted in Figure 5.

### 3.2. Laboratory Data

Blood samples obtained from patients were collected into tubes with or without anticoagulant (with 3.8% sodium citrate) and centrifuged at 300 g for 10 min at room temperature to obtain supernatant. Plasma and serum samples were immediately stored at −80 °C until the time of analysis.

### 3.3. sNOX2-dp, ADMA, and ET-1

Serum NOX2 was measured as sNOX2-dp with an ELISA method as previously reported [23]. Values were expressed as pg/mL; intra- and inter-assay coefficients of variation were 8.95% and 9.01%, respectively.

Quantitative determination of ADMA and ET-1 levels was measured in serum samples by ELISA kit [TEMA ricerca srl, Castenaso (Bologna), Italy] according to the manufacturer’s instructions. The values for ADMA were expressed in ng/mL and for ET-1 in pg/mL. Both intra- and inter-assay coefficients of variation were <10%.

### 3.4. Nitric Oxide (NO) Assay

NO production was evaluated in the serum samples. A colorimetric assay kit (Abcam, Cambridge, UK) was used to determine the metabolites of NO (nitrites and nitrates, NOx) in 100 μL of samples under stirring conditions for 10 min at 37 °C. Values are expressed as µM. Intra- and inter-assay coefficients of variation were 2.9% and 1.7%, respectively.

### 3.5. Radiological Data

Two independent investigators used the ABC/2 formula to calculate the hematoma volume. At CT images, we applied the ABC/2 formula to the hemorrhagic lesion visible as hyperdensity (+30–45 Hounsfield unit, HU). For MRI images, we applied the formula to the visible hypointensity in the GRE sequence, while for the calculation of the edema volume, the score was applied to the visible hyperintensity in the FLAIR sequence. Perihematomal edema volume was measured by subtracting the hematoma volume from the combined hematoma and perihematomal hyperintensity area volumes. Perihematomal edema was considered mild if only sulci effacement was present, moderate in the presence of ventricle asymmetry, and severe if there was a midline shift. The midline shift (mm) was also measured. We also took into consideration the hematoma growth defined as any volume change from T1 to T2. The “island sign”, defined as ≥3 small discrete hematomas that are not connected to the primary hematoma or ≥4 small hematomas that may branch off from the primary hematoma and resemble “bubbles” or “sprouts” rather than a lobulated appearance, was evaluated at the basal brain CT. Finally, we considered mean transit time (MTT), cerebral blood flow (CBF), and cerebral blood volume (CBV), for the evaluation of hypoperfusion related to the hemorrhagic lesion. We measured CBF, CBV, and MTT values in 5 round regions of interest of about 0.5 cm^2^ within 1 cm from the hematoma periphery and drawn freehand on the slice in which the hemorrhage was more visible. Values were averaged and expressed as continuous and categorical variables. According to a previously published paper [6], CBF was categorized into normal (40–55 mL/100 g/min) and low (<40 mL/100 g/min); CBV was dichotomized into normal (>2.5 mL/100 g) and low (≤2.5 mL/100 g) and MTT was dichotomized into normal (≤5 s) and high (>5 s).

According to the results from MRI images, we divided the patients into two groups: patients with the presence of one or more ischemic lesions in the diffusion sequences at T1 and patients without ischemic lesions at T1.

### 3.6. Statistical Analysis

The descriptive statistical analysis of the total population and of the two groups of interest was carried out by calculating means or medians for continuous variables based on their normal or non-normal distribution as well as by calculating frequencies and proportions (or percentages) for dichotomic/categorical variables. For comparison between the two groups of patients with ischemic lesions and patients without ischemic lesions, continuous variables with normal and non-normal distributions were compared using the *t*-test for paired samples or the Mann–Whitney U test, respectively. We compared categorical variables using the Fisher test or chi-square test when appropriate. Correlations between variables were assessed by calculating the Pearson or Spearman coefficient. We presented the course of the molecular biomarkers across the different time points, as well as the median changes over time by the presence/absence of ischemic lesions at T1. Given the small number of patients included in the study, we performed an exploratory multivariate logistic regression to detect potential predictors of the development of perihematomal ischemic lesions after adjustment for age, sex, and variables with a univariate *p*-value < 0.05. For all tests that were performed, a value of *p* < 0.05 was considered statistically significant. Statistical analysis was performed using SPSS statistical software (IBM Corp, SPSS Statistics for Windows, Version 25, Armonk, NY, USA).

## 4. Discussion

In this study, we aimed to identify biological markers potentially involved in the development of secondary brain injury following ICH, specifically focusing on perihematomal hypoperfusion, ischemic lesions, hematoma expansion, and edema development.

The pathophysiology of ICH is still not fully understood. Beyond the initial phase, where extravasated blood causes mechanical disruption of brain structures and increased intracranial pressure, a second phase follows. This phase is characterized by hematoma expansion and the development of vasogenic edema due to the rupture of the BBB [7]. A complex series of events occurs during this phase, involving hemoglobin, iron, hemin, and thrombin toxicity. These factors contribute to the disruption of the endothelium and tight junctions, leading to inflammatory cell invasion, increased calcium and free radical production, extensive microglia activation, and ultimately, increased neuronal death and axonal damage [5].

Our study found that approximately 42% of ICH patients developed ischemic damage within 48–72 h after onset, consistent with previous studies [24]. Patients with ischemic perihematomal lesions had higher sNOX2-dp levels at admission compared to those without lesions. Furthermore, in the exploratory multivariate analysis, high sNOX2-dp values at T0 were associated with the development of perihematomal ischemic lesions at T1, although with a borderline statistical significance likely due to our small sample size. Therefore, these results should be interpreted with caution and need to be confirmed by larger studies. In addition, sNOX2-dp levels were significantly associated with the development of severe edema at 72 h. While no significant correlation was observed between ischemic lesions and worse outcomes as measured by the mRS at three months, we found a significant association between ischemic lesions and any increase in hematoma volume, perilesional edema, and greater midline shift. These findings suggest that oxidative stress, indicated by sNOX2-dp levels, plays a key role in the development of ischemic perihematomal lesions, aligning with previous research indicating that oxidative stress contributes to secondary brain injury in ICH [5,6,7,8,9].

Furthermore, in response to NOX2 activation, we observed a decrease in ET-1 levels. Since the increased formation of superoxide anions by NOX2 activation promotes ET-1 reduction in the endothelium [25], suggesting a putative link between the NOX2 activation and ET-1 expression, we speculated that the reduction in ET-1 levels in the endothelium might represent a compensatory response to modulate and counteract the oxidative stress and endothelial dysfunction observed in this context.

In fact, our results showed an inverse correlation between ET-1 levels at 12–24 h and hemorrhage volume at 48–72 h. This suggests that ET-1 may exert a protective effect by inducing vasoconstriction of the ruptured vessel, limiting hematoma expansion. ET-1 is a powerful vasoconstrictor locally released by the endothelium at the site of vessel injury, initiating vasospasm as the first step in primary hemostasis, potentially leading to the cessation of bleeding [26]. In our study, patients with higher ET-1 plasma levels exhibited less hematoma growth and did not develop ischemic damage. This suggests that ET-1 might induce vasoconstriction in the damaged artery responsible for the bleeding, thereby reducing blood supply to the injured area and preventing hematoma expansion. This reduction in hematoma growth may protect the surrounding parenchyma and prevent the development of ischemic perihematomal lesions. These findings contradict those of Alioğlu et al. [27], who reported that higher ET-1 levels were associated with increased hematoma volume and worse prognosis. To the best of our knowledge, the study by Alioğlu’s group is the only one in the literature that addresses the potential role of ET-1 in patients with ICH. Most research has focused on the role of ET-1 in ischemic stroke and subarachnoid hemorrhage. In ischemic stroke, an acute increase in ET-1 plasma levels has been associated with cerebral vasoconstriction, reduced regional blood flow and microcirculation, and larger infarct volumes [28,29,30]. Additionally, animal studies on subarachnoid hemorrhage have demonstrated that ET-1 administration into cerebrospinal fluid can induce severe vasospasm [31,32,33].

The discrepancy between the findings on ET-1 levels in our study and Alioğlu et al.’s [27] study likely results from a combination of factors, including differences in study design, timing and methodology of ET-1 measurement, patient populations, severity of ICH, clinical outcome measurement tools, and follow-up times. A more detailed exploration of these variables in future studies could help to clarify the role of ET-1 in ICH and resolve these discrepancies. It is also possible that ET-1 exerts different actions depending on the timing of the bleeding onset. The vasoconstrictor effect may be beneficial at the beginning of the bleeding process, but it could become harmful in later phases when reduced blood flow might cause surrounding hypoperfusion and perihematomal damage. Moreover, the endothelin system is very complex, and its effects can vary depending on the receptors activated: stimulation of the ET_A_ receptor induces potent and prolonged vasoconstriction, inflammation, and cell proliferation, whereas stimulation of the ET_B_ receptor generally produces the opposite effects [33].

To confirm the role of vasoconstriction as a compensatory response to oxidative stress conditions, we also evaluated ADMA levels, which are an endogenous inhibitor of NO synthase, and its elevated concentrations in the blood are found in numerous diseases associated with endothelial dysfunction [34]. Moreover, it was demonstrated that ADMA activates the local renin–angiotensin system, and the angiotensin II released activates NOX; superoxide produced interferes with the bioavailability of NO, resulting in diminished flow-induced dilation, a mechanism that may contribute to the development of endothelial dysfunction increasing tone associated with elevated ADMA levels [35]. We found that ADMA levels decreased at T1 and T2 in comparison with T0, and we also found a significant inverse correlation between the levels of ADMA and NIHSS scores at T2. These results corroborate our hypothesis that a reduction in the bioavailability of NO and, consequently, a reduction in NO-mediated dilations are countered by compensatory mechanisms such as a reduction in ET-1 and ADMA levels.

The primary limitation of our study is the small sample size, which may have limited our ability to detect statistically significant differences for some biomarkers. Additionally, the single-center design and the specific geographic location of the cohort may limit the generalizability of our findings to broader, more diverse populations. Finally, the complexity of the study design and the use of surrogate markers for NO production could affect the interpretation of our findings.

The strengths of this study lie in its comprehensive biomarker analysis, integration of advanced imaging techniques, focus on clinically relevant outcomes, exploration of understudied biomarkers, and its prospective design.

In conclusion, our study suggests that NOX2 is linked to the development of ischemic perihematomal lesions, while ET-1 may help to limit hematoma expansion by inducing vasoconstriction of the bleeding artery. Conversely, NO appears to be associated with worse outcomes and increased mortality, likely due to its role in free radical production. Larger, prospective studies are needed to confirm these findings and further clarify the roles of these biomarkers in ICH. Additionally, animal models could be instrumental in providing deeper insights into the underlying mechanisms, ultimately aiding in the development of targeted therapeutic strategies to improve patient outcomes.

## Figures and Tables

**Figure 1 ijms-25-13180-f001:**
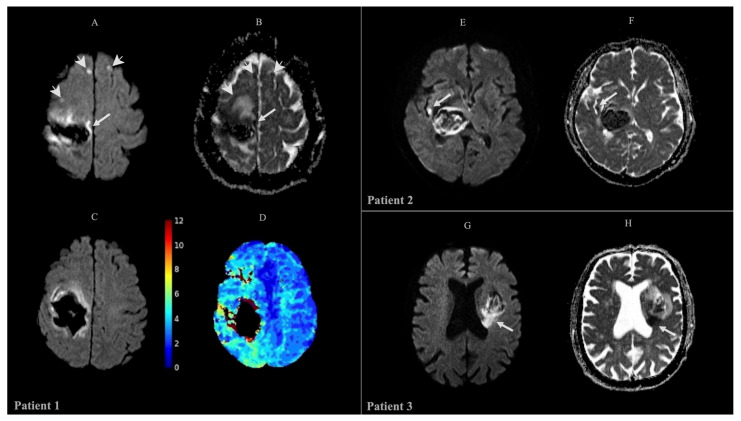
Representative neuroimaging of three patients. Patient 1: MRI at T2 reveals a large intraparenchymal hematoma within the right fronto-parietal lobe. (**A**) DWI sequences identify linear regions of restricted diffusion in the anterior perilesional area and the posteromedial region, with corresponding hypointensity on the ADC map (**B**), suggestive of ischemic injury (arrow). Additionally, small ischemic foci, measuring a few millimeters, are observed in the right frontal subcortical paramedian region and at the left frontal subcortical level (arrowheads). (**D**) Perfusion-weighted imaging (PWI) shows a significant increase in Tmax within the right parieto-occipital region and posterior temporal lobe, surrounding a large hematoma located in the right fronto-parietal lobe (**C**). Patient 2: MRI at T1 demonstrates a hematoma located in the right thalamo-capsular region with slight contralateral shift in the midline structures. (**E**) DWI imaging shows a perihematomal 5 mm^2^ area of diffusion restriction, with corresponding hypointensity on the ADC map (**F**), consistent with a small recent ischemic lesion (arrow). Patient 3: MRI at T1 reveals a hemorrhagic focus in the left insula and basal ganglia, with slight lateral ventricle compression. (**G**) DWI imaging demonstrates a region of restricted diffusion in the posteromedial perilesional area, with corresponding hypointensity on the ADC map (**H**), suggestive of ischemic injury (arrow).

**Figure 2 ijms-25-13180-f002:**
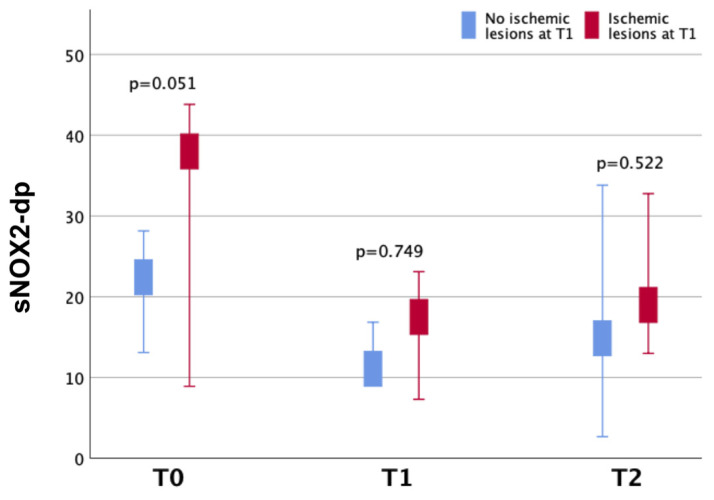
Median plasma levels (pg/mL) of sNOX2-dp at different time points (T0, T1, T2) by presence/absence of ischemic lesions at T1.

**Figure 3 ijms-25-13180-f003:**
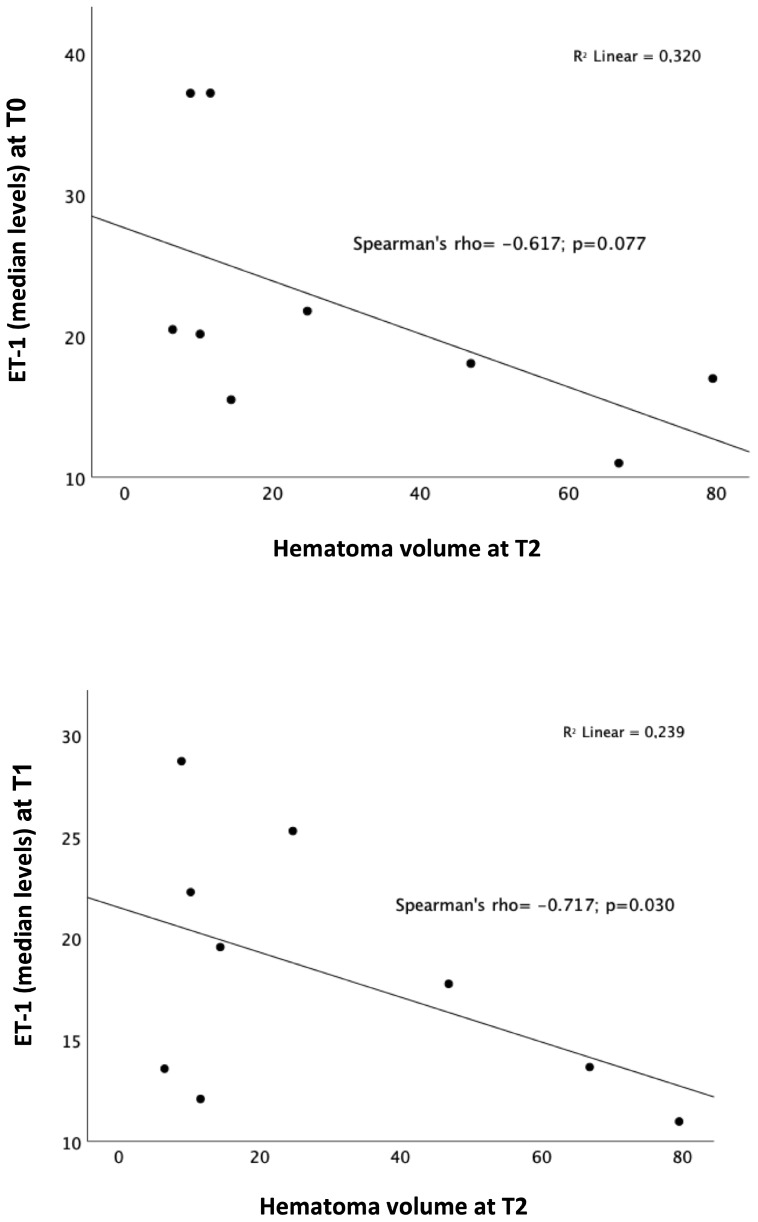
Correlations between ET-1 levels at T0 and T1 and hematoma volume at T2. The statistical test is the Spearman correlation test; black dots represent the single data points; ET-1 levels were measured in pg/mL; hematoma volume was measured in cm^3^.

**Figure 4 ijms-25-13180-f004:**
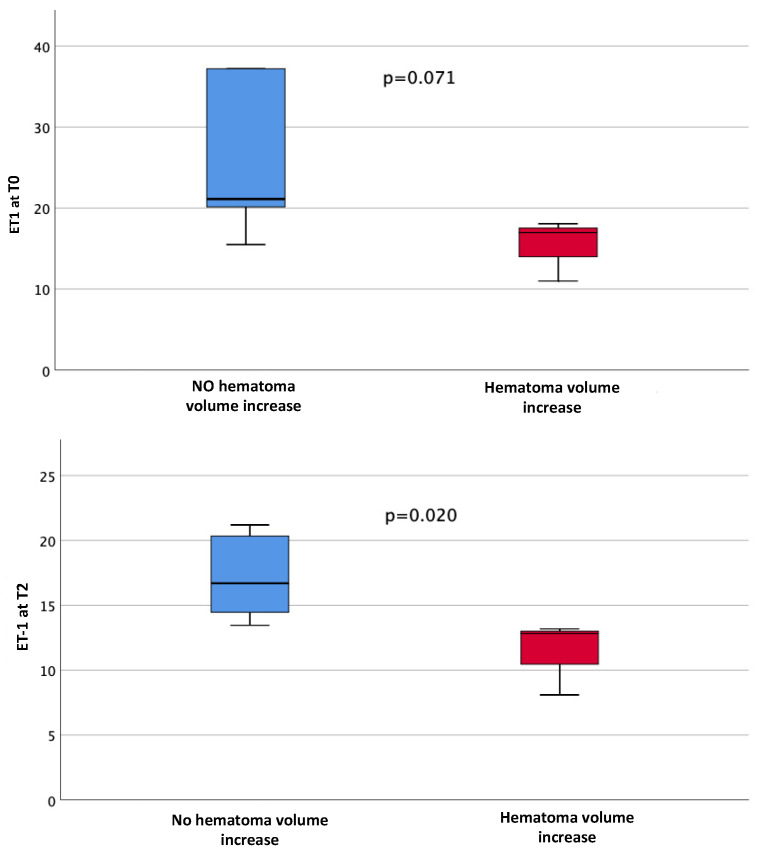
Associations between ET-1 levels (pg/mL) at T0 and T2 and hematoma volume increase from T1 to T2.

**Figure 5 ijms-25-13180-f005:**
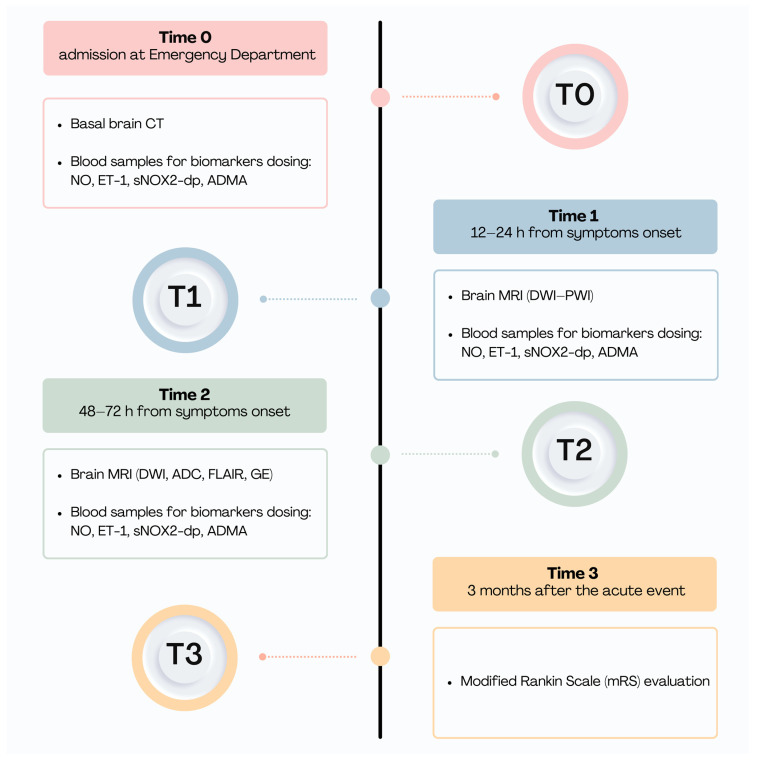
Study timeline.

**Table 1 ijms-25-13180-t001:** Clinical and demographic characteristics of patients according to the presence/absence of ischemic lesions at T1.

	All Patients n = 28	Ischemic Lesions n = 12	No Ischemic Lesions n = 16	*p*
** *Demographic characteristics and clinical history* **				
Age, mean (SD)	70.2 (13.8)	68.4 (15.5)	70.8 (13.0)	0.661
Sex (female) (%)	13 (44.8)	6 (50)	6 (37.5)	0.508
Pre-stroke mRS (%)				0.620
- 0	19/28 (67.9)	7/11 (63.6)	12 (75.0)	
- 1	5/28 (17.9)	2/11 (18.2)	3 (18.8)	
- 2	1/28 (3.6)	0	0	
- 3	3/28 (10.7)	2/11 (18.2)	1 (6.3)	
- 4	0	0	0	
- 5	0	0	0	
Pre-stroke mRS 0–1 (%)	23/27 (85.2)	9/11 (81.8)	14/15 (93.3)	0.556
Smoking (%)				0.174
- no	7 (24.1)	3 (25.0)	4 (25.0)	
- current	15 (51.7)	8 (66.7)	6 (37.5)	
- previous	7 (24.1)	1 (8.3)	6 (37.5)	
Obesity (%)	7 (24.1)	3 (25.0)	4 (25.0)	1.0
Alcohol consumption (%)	2 (6.9)	0	2 (12.5)	0.492
Drug abuse consumption (%)	1 (3.4)	0	1 (6.3)	1.0
Hypertension (%)	22 (75.9)	12 (100.0)	9 (56.3)	**0.010**
Dyslipidemia (%)	10 (34.5)	9 (75.0)	1 (6.3)	**<0.001**
Atrial fibrillation (%)	6 (20.7)	2 (16.7)	3 (18.8)	1.0
Ischemic cardiopathy (%)	4 (13.8)	1 (8.3)	3 (18.8)	0.613
Diabetes mellitus (%)	7 (24.1)	4 (33.3)	3 (18.8)	0.418
Previous stroke (%)	2 (6.9)	0	2 (12.5)	0.492
Previous TIA (%)	0	0	0	-
Coagulopathy (%)	1 (3.4)	0	1 (6.3)	1.0
Valvulopathy (%)	2 (6.9)	0	1 (6.3)	1.0
Peripheral vasculopathy (%)	1 (3.4)	0	1 (6.3)	1.0
Migraine (%)	0	0	0	-
Cancer (%)	2 (6.9)	1 (8.3)	1 (6.3)	1.0
Stressful events (%)	3 (10.3)	1 (8.3)	2 (12.5)	1.0
** *Therapy before hemorrhagic stroke* **				
Anticoagulants (%)	7 (24.1)	2 (16.7)	4 (25.0)	0.673
Antiplatelets (%)	8 (28.6)	4/11 (36.4)	4 (25.0)	0.675
Lipid-lowering medications (%)	2 (6.9)	2 (16.7)	0	0.175
ACE-1/ARB (%)	12 (42.9)	7/11 (63.6)	4 (25.0)	0.061
Diuretics (%)	5 (17.2)	2 (16.7)	2 (12.5)	1.0
Beta-blockers (%)	5 (17.2)	3 (25.0)	2 (12.5)	0.624
Alpha-lytics (%)	4 (13.8)	3 (25.0)	1 (6.3)	0.285
Calcium antagonists (%)	0	0	0	-
PPI (%)	3 (10.3)	1 (8.3)	2 (12.5)	1.0
Blood-glucose-lowering medications (%)	3 (10.3)	2 (16.7)	1 (6.3)	0.560
Insulin (%)	0	0	0	-
Nitrates (%)	0	0	0	-
** *Clinical characteristics of hemorrhagic stroke* **				
Stroke on awakening (%)	11/29 (37.9)	7 (58.4)	4 (25.0)	0.121
NIHSS, median (IQR)				
- Admission (T0)	11 (7.5–18)	14 (6.75–17.50	9 (7.25–18.25)	0.642
- 12–24 h (T1)	9 (6.5–13.5)	13 (6–20)	9 (7.75–10.50)	0.667
- 48–72 h (T2)	10 (7.5–14.5)	14 (6–20)	8.50 (7.75–10.50)	0.197
- Discharge	7 (5–11.5)	8 (4–12.50)	6 (5–11)	0.720
SBP, mean (SD)				
- T0	168.4 (31.50)	168.5 (30.3)	168.9 (34.4)	0.977
- T1	145.8 (25.9)	150.1 (24.3)	140.7 (29.0)	0.534
- T2	147.2 (22.4)	149.9 (13.9)	143.4 (32.7)	0.646
DBP, mean (SD)				
- T0	92.57 (19.21)	91.4 (20.1)	94.0 (19.7)	0.740
- T1	85.8 (17.6)	88.9 (16.1)	81 (19.4)	0.387
- T2	84.1 (16.7)	83.3 (10.0)	85.2 (24.8)	0.855
HR, mean (SD)				
- T0	80.6 (13.71)	78.6 (15.5)	81.1 (12.6)	0.652
- T1	81.2 (21.2)	88 (22.6)	73.3 (18.1)	0.228
- T2	79.3 (12.3)	83.9 (19.6)	73.0 (10.0)	0.362
Body temperature, mean (SD)				
- T0	36.0 (0)	36 (0)	36.0 (0)	-
- T1	36.9 (0.70)	27.2 (0.8)	36.6 (0.3)	0.108
- T2	36.9 (0.6)	37.2 (0.4)	36.6 (0.6)	**0.043**
HGT, mean (SD)				
- T0	128 (29.4)	136 (32.9)	120 (24.6)	0.143
- T1	128 (32.7)	128 (30.9)	130 (38.7)	0.949
- T2	119 (23.0)	118 (25.8)	120 (97.5–137.5)	0.900
Craniotomy (%)	3/19 (15.8)	2/8 (25.0)	1/11 (9.1)	0.546

SD = standard deviation; ACE-1/ARBs = angiotensin-converting enzyme inhibitors/angiotensin II receptor blockers; PPIs = proton pump inhibitors; NIHSS = National Institutes of Health Stroke Scale; IQR = interquartile range; SBP = systolic blood pressure; DBP = diastolic blood pressure; HR = heart rate; HGT = hemo-glucose test. *p*-values highlighted in bold indicate statistically significant results (*p* < 0.05).

**Table 2 ijms-25-13180-t002:** Blood test results of patients according to the presence/absence of ischemic lesions at T1.

	All Patients n = 28	Ischemic Lesions n = 12	No Ischemic Lesions n = 16	*p*
** *Routine blood tests* **				
Time onset, blood sampling T0, h, median (IQR)	2.08 (1.27–2.57)	2.33 (1.24–4.19)	2.07 (1.14–2.41)	0.429
Time onset T1, h, median (IQR)	26.50 (25.30–27.30)	26.5 (25.6–29.5)	26.60 (25.10, 27.58)	0.927
Time onset T2, h median (IQR)	51.0 (48.5–53.2)	51 (50.25–54.60)	49.9 (48.3–52.9)	0.361
Haematocrit, T0, mean (SD)	40.5 (5.1)	38.7 (5.2)	42.2 (4.7)	0.073
Platelets, T0, mean (SD)	232.5 (74.95)	233.3 (67.3)	222.8 (75.2)	0.703
**Total leukocytes, mean (SD)**				
**- T0**	8.66 (2.96)	10.18 (3.31)	7.42 (2.17)	**0.013**
- T1	10.51 (5.24)	12.32 (5.90)	7.50 (2.21)	0.234
- T2	11.83 (7.87)	13.52 (9.40)	8.46 (1.32)	0.400
**Neutrophils (absolute number),**				
**mean (SD)**				
**- T0**	6.13 (2.91)	7.44 (3.26)	4.49 (2.35)	**0.037**
- T1	8.46 (5.11)	10.2 (5.83)	5.64 (2.06)	0.254
- T2	9.94 (7.36)	11.55 (8.76)	6.71 (1.14)	0.387
Neutrophils (%)				
- T0	68.90 (12.94)	71.20 (11.88)	67.4 (14.2)	0.456
- T1	77.56 (8.06)	79.62 (9.25)	74.13 (5.26)	0.392
- T2	81.86 (5.26)	83.23 (6.03)	79.10 (1.66)	0.295
Lymphocytes (absolute number),				
mean (SD)				
- T0	1.84 (1.03)	1.94 (1.0)	1.70 (1.08)	0.555
- T1	77.56 (8.06)	1.31 (0.62)	1.11 (0.19)	0.612
- T2	1.08 (0.35)	1.09 (0.43)	1.06 (0.16)	0.912
Lymphocytes (%), mean (SD)				
- T0	22.40 (11.35)	20.24 (9.71)	23.69 (12.81)	0.443
- T1	13.60 (5.95)	12.68 (7.52)	15.13 (2.29)	0.612
- T2	10.41 (3.20)	9.23 (3.36)	12.77 (0.56)	0.123
CRP (mg/dL), mean (SD)				
- T0	0.51 (0.72)	0.79 (0.97)	0.23 (0.24)	0.111
- T1	2.33 (1.68)	3.10 (1.59)	1.06 (0.95)	0.094
- T2	6.17 (6.19)	8.70 (6.75)	1.94 (0.07)	0.144
INR, T0, mean (SD)	1.09 (0.22)	1.04 (0.10)	1.13 (0.28)	0.253
PTT ratio, T0, mean (SD)	0.96 (0.23)	0.94 (1.76)	0.97 (0.27)	0.804
Fibrinogen (md/dL), T0, mean (SD)	374.38 (90.50)	389.3 (84.5)	358.81 (96.37)	0.391
Total cholesterol (mg/dL), T0, mean (SD)	162.63 (32.02)	161.0 (21.6)	164.3 (43.8)	0.898
LDL (md/dL), T0, mean (SD)	103.25 (23.40)	109.5 (19.7)	97.0 (28.0)	0.493

IQR = interquartile range; SD = standard deviation; CRP = C-reactive protein; INR = International Normalized Ratio; PTT = prothrombin time test; LDL = low-density lipoproteins. *p*-values highlighted in bold indicate statistically significant results (*p* < 0.05).

## Data Availability

Data are contained within the article and Appendix A.

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
