# Peer review of "Secondary Brain Injury After Parenchymal Cerebral Hemorrhage in Humans: The Role of NOX2-Mediated Oxidative Stress and Endothelin-1"

_ijms, 2024, doi:10.3390/ijms252313180_

Round 1
Reviewer 1 Report
Comments and Suggestions for Authors
Comments to the Manuscript “ijms-3310063” entitled “Secondary brain injury after parenchymal cerebral hemorrhage in humans: the role of NOX2-mediated oxidative stress and Endothelin-1” by De Michele M. et al.
General comment.
The manuscript by White A. L. et al., is a study of serum biomarkers related to inflammation and oxidative stress, and vasoactive molecules in model of hypoperfusion and ischemic perihematomal lesions following intraparenchymal cerebral hemorrhage (ICH), and a comparative analysis of these biomarkers with volume growth of hematoma and perihematomal edema. The authors collected blood samples from 28 patients affected by ICH, at three timepoints based on symptoms onset and hospitalization, and used those samples to measure the expression and production levels of Endothelin-1 (ET-1), nitrites/nitrates (NO), soluble NOX2-derived peptide (sNOX2-dp), and asymmetric dimethylarginine (ADMA). The authors found a positive correlation between the level of the biomarkers and the development of symptoms and growth in lesion volume. The authors then conclude that NOX2 activation may have a role in the development of ischemic perihematomal lesions, and the association between higher ET-1 values and a lower hemorrhage volume could be related to the ET-1 vasoconstriction action on the ruptured vessel wall.
Overall, the manuscript is well written and does not show major concerns in the use of language and grammar. The results are comprehensible and support the discussion.
However, the size of the population used in this study is not too big and some results need a better and deeper analysis using a larger population. Also, the increased expression of some biomarkers may be due to the age of the population used in the study, because it is known that oxidative stress is associated with aging and increased risk of age-related diseases. Moreover, it would be very interesting to understand whether the selected biomarkers are a cause or a consequence of severe symptoms following ICH, as well as they could be used as predictive or diagnostic biomarkers, thus helping in the risk stratification of patients.
Specific major/minor comments.
‒ Page 2 and others: uniform ET1 to ET-1.
‒ Page 2: please extend the abbreviation for ADC.
‒ Figure 1: it could be helpful to add the name of analyzed biomarkers and vasoactive substances between square in the related point at T0 and T1.
‒ Page 4: please uniform intra-assay and inter-assay as in previous points, or viceversa.
‒ Page 4: subsections 2.3, 2.5, and 2.6 can be summarized in a single point, just explaining the difference of the variation coefficient for each molecule.
‒ Figure 2: the text describing figure 2 should be included as caption to the figure, not as separated text. Also, despite each panel represents a different patient, it would be more appropriate to use consequential letters for each panel (A to H), instead of using only A and B. The authors should describe what the arrows are pointing to. Panel B of patient 1 needs an improved image quality. Also, the authors might include PWI images for both patients 2 and 3.
‒ Subsection 3.4: the authors should include at least a representative figure for this section, as well as for section 3.3, since they include these as separate result sections. Moreover, please uniform sNOX2-dp in Supp. Figure 1, because the y axis describes it as NOX2. Also, the "borderline statistical significance" might be a random expression or only a predictive biomarker of ischemic lesion, since it is completely ablated at T1 and T2 between the 2 groups. The authors could discuss about it.
‒ Subsection 3.5: please redo Figure 3, removing Italian axis, uniforming text, including caption, describing black dots, including unit of measurement for hematoma volume and ET-1 level, describing the method and comparison used for statistical significance.
‒ Page 10: the authors claim there is no significant association between levels of biomarkers in this section, but actually they show a statistically significant change in Supp. Figure 7. Could this be discussed?
‒ Supp. Figure 6: the graph is partially covered by a NIHSS at T2 annotation, please remove it.
‒ The authors should include a caption for all supplementary material, both tables and figures.
Author Response
Reviewer 1.
General comment.
The manuscript by De Michele et al., is a study of serum biomarkers related to inflammation and oxidative stress, and vasoactive molecules in model of hypoperfusion and ischemic perihematomal lesions following intraparenchymal cerebral hemorrhage (ICH), and a comparative analysis of these biomarkers with volume growth of hematoma and perihematomal edema. The authors collected blood samples from 28 patients affected by ICH, at three timepoints based on symptoms onset and hospitalization, and used those samples to measure the expression and production levels of Endothelin-1 (ET-1), nitrites/nitrates (NO), soluble NOX2-derived peptide (sNOX2-dp), and asymmetric dimethylarginine (ADMA). The authors found a positive correlation between the level of the biomarkers and the development of symptoms and growth in lesion volume. The authors then conclude that NOX2 activation may have a role in the development of ischemic perihematomal lesions, and the association between higher ET-1 values and a lower hemorrhage volume could be related to the ET-1 vasoconstriction action on the ruptured vessel wall.
Overall, the manuscript is well written and does not show major concerns in the use of language and grammar. The results are comprehensible and support the discussion.
However, the size of the population used in this study is not too big and some results need a better and deeper analysis using a larger population. Also, the increased expression of some biomarkers may be due to the age of the population used in the study, because it is known that oxidative stress is associated with aging and increased risk of age-related diseases. Moreover, it would be very interesting to understand whether the selected biomarkers are a cause or a consequence of severe symptoms following ICH, as well as they could be used as predictive or diagnostic biomarkers, thus helping in the risk stratification of patients.
We thank the reviewer for these insightful comments. We acknowledge that a key limitation of the present study is the small sample size, which we have addressed in the discussion. However, the study design is particularly complex, especially as it was conducted in the acute and subacute phases of stroke, within the Emergency Department setting.
Oxidative stress is associated with aging and an increased risk of age-related diseases. However, we did not find any association between age and the plasma levels of oxidative stress markers. The two groups examined (patients with and without ischemic lesions) were homogeneous in terms of age.
Although it is difficult to determine whether the selected biomarkers are a cause or consequence of severe symptoms following ICH, we believe that higher NO levels at T0 could be a cause of the severity of symptoms at discharge rather than a consequence. In fact, we did not find any association at the earlier time points between NO levels and the NIHSS score, suggesting that NO could be used as a predictive prognostic biomarker.
We apologize for the error in the main text, which we have now corrected. On page 10, we replaced "T1" with "discharge”: “A significant linear positive correlation was found between high levels of some biomarkers and neurological severity measured by NIHSS score at different timepoints, such as NO at T1 (Spearman’s rho=0.783, p=0.007) with NIHSS at T1 discharge...”
Specific major/minor comments.
‒ Page 2 and others: uniform ET1 to ET-1. Done
‒ Page 2: please extend the abbreviation for ADC.
We have written out the acronym ADC in full as Apparent Diffusion Coefficient (ADC) map.
‒ Figure 1: it could be helpful to add the name of analyzed biomarkers and vasoactive substances between square in the related point at T0 and T1. Done
‒ Page 4: please uniform intra-assay and inter-assay as in previous points, or viceversa. We have uniformed to intra- and inter-assay
‒ Page 4: subsections 2.3, 2.5, and 2.6 can be summarized in a single point, just explaining the difference of the variation coefficient for each molecule.
As suggested, subsections 2.5 and 2.6 have been included in the subsection 2.3.
‒ Figure 2: the text describing figure 2 should be included as caption to the figure, not as separated text. Also, despite each panel represents a different patient, it would be more appropriate to use consequential letters for each panel (A to H), instead of using only A and B. The authors should describe what the arrows are pointing to. Panel B of patient 1 needs an improved image quality. Also, the authors might include PWI images for both patients 2 and 3.
We thank the reviewer for these valuable suggestions, which have helped improve the clarity of the paper’s results. In response, we have removed the patient's description from the main text and instead included it in the caption of Figure 2. Additionally, we have added labels (A to H) to each panel for better organization. The arrows' descriptions have been updated to specify what they are pointing to, and we have modified the shape of the arrows to enhance clarity. Lastly, considering that the PWI images are quite similar across the three patients, we opted not to include the PWI images for patients 2 and 3 in order to avoid making the figure potentially confusing. We have also included the full form of the acronyms at the end of the figure captions for clarity.
‒ Subsection 3.4: the authors should include at least a representative figure for this section, as well as for section 3.3, since they include these as separate result sections. Moreover, please uniform sNOX2-dp in Supp. Figure 1, because the y axis describes it as NOX2. Also, the "borderline statistical significance" might be a random expression or only a predictive biomarker of ischemic lesion, since it is completely ablated at T1 and T2 between the 2 groups. The authors could discuss about it.
We thank the reviewer for these comments. Regarding section 3.3, since no statistically significant between-group differences were found in terms of functional outcome at 3 months, intra-hospital death, and mortality at 3 months, as indicated by the Supplementary Table 2, we think that moving the Supplementary Table 2 to the main text or adding representative graphs on non-statistically significant results to the main text could be confounding for the readers.
Regarding section 3.4, we have moved the more representative graphs on borderline significant results, namely those regarding the difference in the serum concentration of sNOX2-dp at T0 between patients with ischemic lesions at T1 and those without. We left the other non-statistically significant results in the Supplementary Tables in order to avoid too many tables and Figures in the main text.
We agree with the reviewer that the serum concentrations of sNOX2-dp at admission could represent a potential independent predictor of development of perihematomal ischemic lesion in patients with ICH, since the borderline statistically significant differences between patients with and without ischemic lesions at univariate analysis could be due the small sample size. This was evident also from the exploratory multivariate analysis, where after adjustment for age, sex, neutrophil absolute number at T0 and hematoma volume at T1, high sNOX2-dp values at T0 were independently associated with the development of perihematomal ischemic lesions at T1, although with a trend toward the statistical significance (OR 1.122, 95% CI 0.991-1.269, p: 0.068). We strongly believe that in larger study the nominal statistical significance could be reached. We have further elaborated on these aspects in the Discussion as follows:
“Our study found that approximately 42% of ICH patients developed ischemic damage within 48-72 hours after onset, consistent with previous studies 24. Although Patients with ischemic perihematomal lesions had higher sNOX2-dp levels at admission compared to those without lesions. Furthermore, in the exploratory multivariate analysis high sNOX2-dp values at T0 were associated with the development of perihematomal ischemic lesions at T1, although with a borderline statistical significance likely due to our small sample size. Therefore, these results should be interpreted with caution and need to be confirmed by larger studies. However, In addition, sNOX2-dp levels were significantly associated with the development of severe edema at 72 hours…”
‒ Subsection 3.5: please redo Figure 3, removing Italian axis, uniforming text, including caption, describing black dots, including unit of measurement for hematoma volume and ET-1 level, describing the method and comparison used for statistical significance.
We apologize for these mistakes. We have now removed from Figure 4 the Italian axis, included caption reporting the method of comparison used for statistical significance, describing black dots, and unit of measurement for ET-1 levels and hematoma volume. We have corrected the Figure 4 legend as follows:
“Figure 4. Correlations between molecular biomarkers ET-1 levels at T0 and T1 and hematoma volume at T2.
The statistical test is the Spermann correlation test; black dots represent the single data points; ET-1 levels were measured in pg/mL; hematoma volume was measured in cm3.”
We have also revised the Figure 3 and 5 legends by introducing the unit of measurement of sNOX2-dp and ET-1 as follows:
“Figure 3. Median plasma levels (pg/mL) of sNOX2-dp at different timepoints (T0, T1, T2) by presence/absence of ischemic lesions at T1.”
“Figure 5 Associations between molecular biomarkers ET-1 levels (pg/mL) at T0 and T2 and hematoma volume increase from T1 to T2.”
‒ Page 10: the authors claim there is no significant association between levels of biomarkers in this section, but actually they show a statistically significant change in Supp. Figure 7. Could this be discussed?
We thank the reviewer for this observation. In the Supplementary Figures we included all those associations with a borderline statistical significance or a trend towards the statistical significance. However, in the Results of the main text we tried to mostly include those associations with a clear nominal statistical significance in order to avoid confusion in the readers. To avoid misunderstanding we have now removed from the Supplementary figure legends “Statistically significant …” and have left only “Associations …”
‒ Supp. Figure 6: the graph is partially covered by a NIHSS at T2 annotation, please remove it. Done
‒ The authors should include a caption for all supplementary material, both tables and figures.
We have tried to revise these aspects in the Supplementary material. We had included titles to supplementary tables and legends to the supplementary figures; we have now reported unit of measurements and abbreviations in the tables and in the tables associated to the figures, the definitions of some variables (e.g., the severity of edema, hypoperfusion, hematoma growth, outcome measures) are reported in the Methods of the main text.
Reviewer 2 Report
Comments and Suggestions for Authors
The article examines the role of oxidative stress and vasoactive substances in secondary brain injury following ICH. The study addresses a clinically significant topic, however, certain issues need to be deeper clarified and fixed.
These limitations should be better clarified: with only 28 participants, the small cohort size impairs the statistical power to detect meaningful differences, particularly regarding biomarker levels and clinical outcomes. Also, the study cohort includes patients from a single institution and a specific geographic location. This restricts the external validity of the findings, as the patient population may not represent broader demographic or clinical variations.
In addition, although the study adjusts for some factors, such as age and sex, it lacks comprehensive control for other potential confounders, including pre-existing comorbidities or concurrent treatments. Why did authors not perform a more thorough multivariate analysis to isolate the effects of the studied biomarkers?
Several reported associations, such as those involving sNOX2-dp levels and ischemic lesions, approach but do not achieve statistical significance. The exploratory multivariate regression results, with p-values like 0.068, underscore the need for cautious interpretation. A clearer distinction between exploratory and confirmatory analyses would improve the study's understanding for the readers.
While the study highlights changes in biomarkers over time, it does not sufficiently discuss their physiological implications or causative relationships with observed clinical outcomes. Did the authors think a more mechanistic exploration, (for instance, supported by animal models or additional experimental data) would provide a deeper understanding of the processes involved?
Although the study correlates biomarkers with radiological findings, it fails to demonstrate a robust link between these biomarkers and long-term functional outcomes such as mortality or disability. Is this gap reducing the study's translational potential for clinical practice?
Author Response
Reviewer 2.
The article examines the role of oxidative stress and vasoactive substances in secondary brain injury following ICH. The study addresses a clinically significant topic, however, certain issues need to be deeper clarified and fixed.
These limitations should be better clarified: with only 28 participants, the small cohort size impairs the statistical power to detect meaningful differences, particularly regarding biomarker levels and clinical outcomes. Also, the study cohort includes patients from a single institution and a specific geographic location. This restricts the external validity of the findings, as the patient population may not represent broader demographic or clinical variations.
We appreciate the reviewer’s comment regarding the limitations of our study, particularly the small sample size and the cohort being from a single institution. We agree that the sample size of 28 participants limits the statistical power, and we have made this point clear in the discussion section of the manuscript. We acknowledge that the small cohort size may impact the ability to detect statistically significant differences, especially in terms of biomarker levels and clinical outcomes.
Regarding the single-center design and specific geographic location, we recognize that this limits the generalizability of our findings. However, we believe that the data provide valuable insights, and we have highlighted these limitations in the discussion to temper the broader applicability of the results. Future studies with larger, multi-center cohorts are needed to confirm these findings and assess their generalizability to more diverse populations.
We have added the following sentence in the discussion: “Additionally, the single-center design and the specific geographic location of the cohort may limit the generalizability of our findings to broader, more diverse populations”.
In addition, although the study adjusts for some factors, such as age and sex, it lacks comprehensive control for other potential confounders, including pre-existing comorbidities or concurrent treatments. Why did authors not perform a more thorough multivariate analysis to isolate the effects of the studied biomarkers?
We thank the reviewer for this comment. We included age, sex, neutrophil absolute number at T0 and hematoma volume at T1, sNOX2-dp values at T0 in the exploratory multivariate model. We agree that there are several confounders that should be included in a multivariate analysis in a study like ours investigating the association between the levels of peripheral biomarkers and the development of perihematomal ischemic injury in patients with ICH. Unfortunately, we could not adjust for additional variables in order to avoid a model overfitting due to the small sample size which could make more difficult the interpretation of the results. This is the reason for which this multivariate analysis was exploratory and larger studies are needed to confirm the results of our study.
Several reported associations, such as those involving sNOX2-dp levels and ischemic lesions, approach but do not achieve statistical significance. The exploratory multivariate regression results, with p-values like 0.068, underscore the need for cautious interpretation. A clearer distinction between exploratory and confirmatory analyses would improve the study's understanding for the readers.
As we pointed out in the manuscript, most of the borderline statistical significances or the trends towards the statistical significance which we observed in our study results are due to the small sample size. As reported in the response to the previous reviewer’s comment, we agree that the results of the multivariate analysis should be interpreted with caution because the small sample size did not allow us to adjust the model for other important variables that could influence the risk of developing ischemic brain injury in the perihematomal area in patients with ICH. This is the reason for which we considered the multivariate analysis as exploratory because only larger and, possibly, multicenter studies could confirm the results of our study.
We had reported in the Statistical analysis section as follows: “Given the small numbers of patients included in the study, we performed an exploratory multivariate logistic regression to detect potential predictors of the development of perihematomal ischemic lesions after adjustment for age, sex and variables with a univariate p value <0.05”.
Furthermore, in the Discussion we have added: “Patients with ischemic perihematomal lesions had higher sNOX2-dp levels at admission compared to those without lesions. Furthermore, in the exploratory multivariate analysis high sNOX2-dp values at T0 were associated with the development of perihematomal ischemic lesions at T1, although with a borderline statistical significance likely due to our small sample size. Therefore, these results should be interpreted with caution and need to be confirmed by larger studies.”
While the study highlights changes in biomarkers over time, it does not sufficiently discuss their physiological implications or causative relationships with observed clinical outcomes. Did the authors think a more mechanistic exploration, (for instance, supported by animal models or additional experimental data) would provide a deeper understanding of the processes involved?
We thank the reviewer for this comment. We acknowledge that a more mechanistic exploration would deepen our understanding of the underlying processes. However, given the observational nature of our study, we were limited to clinical data and biomarker measurements. We agree that future studies, particularly those involving animal models or additional experimental data, would be beneficial for elucidating the mechanistic pathways and confirming the causal relationships between these biomarkers and clinical outcomes. We have noted this as a potential avenue for future research in the discussion. We change the last sentence of the discussion: “Larger, prospective studies are needed to confirm these findings and further clarify the roles of these biomarkers in ICH. Additionally, animal models could be instrumental in providing deeper insights into the underlying mechanisms, ultimately aiding in the development of targeted therapeutic strategies to improve patient outcomes”.
Although the study correlates biomarkers with radiological findings, it fails to demonstrate a robust link between these biomarkers and long-term functional outcomes such as mortality or disability. Is this gap reducing the study's translational potential for clinical practice?
We appreciate the reviewer’s observation regarding the lack of a robust link between biomarkers and long-term functional outcomes such as mortality or disability. We believe that this gap is likely due to the small sample size of the cohort analyzed, which may have limited the statistical power to detect significant associations between the biomarkers and long-term outcomes. Despite this limitation, we believe that our study offers important preliminary data on the temporal changes of these biomarkers, and larger, more comprehensive studies are needed to fully explore their potential for predicting long-term functional outcomes and improving clinical practice.
Reviewer 3 Report
Comments and Suggestions for Authors
In the work by De Michele et al., Secondary brain injury after parenchymal cerebral hemorrhage in humans: the role of NOX2-mediated oxidative stress and Endotheline-1, the authors observe the role of vasoactive substances (ET-1, NO, and ADMA), markers of oxidative stress (sNOX2-dp) in the development of perihematomal hypoperfusion, ischemic lesions, evolution of hematoma and perilesional edema volume and their relation to clinical outcome.
Main comments:
The authors discuss a large amount of data over 3 months, which results in some lack of clarity.
I have several small comments about the work:
1. I recommend explaining all the abbreviations used when they are used for the first time, similar to the text for tables and figures.
2. I recommend standardizing the entire work and in the supplements the units of mL, pg/mL instead of ml, pg/ml, etc.
3. In tables 1 and 2, list the markers in SI units, e.g. Fibrinogen (g/L), total cholesterol, and LDL cholesterol (mmol/L)
4. Check the number of Lymphocytes (absolute amount) in All patients at T1
5. dysfunction except for disfunction
Author Response
Reviewer 3.
In the work by De Michele et al., Secondary brain injury after parenchymal cerebral hemorrhage in humans: the role of NOX2-mediated oxidative stress and Endotheline-1, the authors observe the role of vasoactive substances (ET-1, NO, and ADMA), markers of oxidative stress (sNOX2-dp) in the development of perihematomal hypoperfusion, ischemic lesions, evolution of hematoma and perilesional edema volume and their relation to clinical outcome.
Main comments:
The authors discuss a large amount of data over 3 months, which results in some lack of clarity.
We appreciate the reviewer’s feedback regarding the clarity of the data presentation. While we understand the concern, we believe that the current use of tables and graphs is the best way to present the large amount of data collected over the 3-month period. We have made efforts to ensure that each table and figure is clearly labeled and that key findings are summarized concisely. Given the complexity and the comprehensive nature of the dataset, further simplification may risk omitting important details that are crucial to the study's findings. We trust that the current presentation provides a balanced approach, but we are open to any specific suggestions the reviewer may have for improving clarity.
I have several small comments about the work:
- I recommend explaining all the abbreviations used when they are used for the first time, similar to the text for tables and figures. We thank the reviewer for addressing this point. We have carefully reviewed the manuscript and standardized the use of abbreviations throughout.
- I recommend standardizing the entire work and in the supplements the units of mL, pg/mL instead of ml, pg/ml, etc. Done
- In tables 1 and 2, list the markers in SI units, e.g. Fibrinogen (g/L), total cholesterol, and LDL cholesterol (mmol/L). Done
- Check the number of Lymphocytes (absolute amount) in All patients at T1. Done
- dysfunction except for disfunction. We apologize for this mistake and have corrected the spelling of the word
Round 2
Reviewer 3 Report
Comments and Suggestions for Authors
The authors accepted the reviewer's recommendations.
Author Response
We thank the reviewer for his suggestions and for collaborating to improve the quality of our research.